# Heparan Sulfate: A Regulator of White Adipocyte Differentiation and of Vascular/Adipocyte Interactions

**DOI:** 10.3390/biomedicines10092115

**Published:** 2022-08-29

**Authors:** J. Michael Sorrell, Arnold I. Caplan

**Affiliations:** Skeletal Biology Research Center, Case Western Reserve University, Cleveland, OH 44106, USA

**Keywords:** white adipose tissue, heparan sulfate, sulfation, adipokines, tissue engineering

## Abstract

White adipose tissues are major endocrine organs that release factors, termed adipokines, which affect other major organ systems. The development and functions of adipose tissues depend largely upon the glycosaminoglycan heparan sulfate. Heparan sulfate proteoglycans (HSPGs) surround both adipocytes and vascular structures and facilitate the communication between these two components. This communication mediates the continued export of adipokines from adipose tissues. Heparan sulfates regulate cellular physiology and communication through a sulfation code that ionically interacts with heparan-binding regions on a select set of proteins. Many of these proteins are growth factors and chemokines that regulate tissue function and inflammation. Cells regulate heparan sulfate sulfation through the release of heparanases and sulfatases. It is now possible to tissue engineer vascularized adipose tissues that express heparan sulfate proteoglycans. This makes it possible to use these tissue constructs to study the role of heparan sulfates in the regulation of adipokine production and release. It is possible to regulate the production of heparanases and sulfatases in order to fine-tune experimental studies.

## 1. Introduction

White adipose tissues (WATs) regulate energy homeostasis through their ability to store and release lipids [1]. Further, these tissues provide thermal insulation and act as shock absorbers [2]. Another critical function is to regulate activities in other organs such as the liver, lungs, heart, and skeletal muscle through the release of bioactive factors termed adipokines, making WAT the largest endocrine organ in the body [3,4]. Adipokines are a highly heterogeneous population of molecules that perform many and varied functions. They include hormones, cytokines, enzymes, growth factors, and chemokines [5]. The endocrine function of WATs depends upon the close and extensive interrelationship between adipocytes and the microvasculature [6]. The development of WATs and their endocrine functions throughout life depends in major part on the presence of heparan sulfate proteoglycans on both adipocyte surfaces and vascular surfaces. This review will examine the functions of HSPGs in these roles. Further, the review will propose the application of engineered vascularized tissue constructed to understand these functions better.

## 2. In Vivo Development of WAT

WAT is found in three anatomic locations: subcutaneous, visceral, and medullary/bone marrow [1,7,8,9,10]. Each of these regions initially develops from microvascular plexi that are supported by mesenchymal mural cells. A subset of these mural cells migrates from the vascular wall and begin to acquire small cytoplasmic lipid deposits [8]. Simultaneously, these cells begin to secrete laminins and type IV collagen, both components of basement membranes (BMs) [7]. They also begin to acquire cell surface receptors for laminin, which allows them to initiate BM organization [11]. Another early cell surface antigen is CD146, also known as melanoma cell adhesion molecule [12]. Tang’s group used genetically marked mice to locate and isolate adipogenic progenitors in WAT [13]. They located previously committed cells in the vascular mural compartment of WAT but not in other tissues. These cells were identified by their expression of the platelet-derived growth factor-β receptor.

The development of a BM around adipocytes is an anomaly since BMs are typically found at sites of epithelial-mesenchymal interfaces where many of the molecular components are secreted by mesenchymal cells. However, the assembly of the BM is driven by laminin receptors on epithelial cell surfaces [14]. Adipocytes are unique mesenchymal cells in that they both produce BM components and express cell surface receptors for laminins, which means that they can organize BMs [15]. This is a critical step in the development of the adipocyte as it provides these cells with both mechanical support and a BM that contains heparan sulfate proteoglycans (HSPGs). Thus, adipose tissues contain two distinct sets of BMs, those that surround adipocytes and those that surround the microvasculature [16].

## 3. Heparan Sulfate Proteoglycans

Each HSPG consists of a core protein to which one or more heparan sulfate (HS) chains are covalently attached [17,18] (Figure 1A,B). Adipose tissues, like other tissues and organs, contain multiple species of HSPGs that appear on both cell surfaces and in the matrix [17,18,19]. These have been allocated into full-time and part-time entities [18] (Table 1), and both sets are present in adipose tissues. It is important to note that adipose depots in various anatomic sites differ physiologically [20]. Gesta and co-workers [20] examined these various sites in both mice and humans to determine whether specific sets of genes might be differentially expressed. They found that a specific set of genes was more highly expressed in mouse intraabdominal epididymal depots compared to other anatomic sites. One of these highly over-expressed genes was the HSPG glypican 4. The same gene was also highly over-expressed in human visceral fat depots, except in lean individuals where it was over-expressed in subcutaneous regions. Glypican 4, like other glypicans, is glycosyl-phosphatidylinositol–anchored (GPI-anchored) to the cell membrane [18]. This allows it to migrate within the plane of the membrane, a feature that enables it to become concentrated at specific sites in the membrane. The part-time HSPG CD44 is also present in adipose tissue. CD44 plays a role in adipose tissue physiology, but it is not clear whether the HS chain is involved [21]. Betaglycan, which functions as an alternate TGFβ receptor, has been shown to be downregulated in the visceral fat of obese individuals [22]. The BM HSPG perlecan is also a major component of adipose tissues. Perlecan is a matrix PG, and its core protein contains multiple binding domains for other matrix molecules [23]. This feature enables its insertion into BMs. It also may interact with cell surfaces via the α2β1 integrin [24].

## 4. Sulfation of Heparan Sulfates and Its Role in Adipose Tissues

The carbohydrate component of HSs consists of repeating disaccharides, each containing an amino sugar and either a uronic or iduronic acid [25]. These are linked into straight chains that are modified during synthesis. Internal cell-mediated modifications include acetylation/deacetylation, epimerization, and sulfation [25]. Sulfation is organized by a family of enzymes termed sulfotransferases [26]. Sulfotransferases are membrane-bound enzymes that are aligned in linear arrays in the Golgi so that they can organize the attachment of sulfate groups to specific regions of the HS chains. There are a maximum of four possible sulfation sites on disaccharides that contain iduronic acid and three possible sites if uronic acid is present (Figure 1B). This means that a given HS disaccharide can have anywhere between 0 and 4 sulfate units. As sulfation is non-random, this produces GAG chains that have distinct domains that recognize heparan binding domains on select proteins [27,28,29,30]. These domains consist of five or more highly sulfated saccharide units. There are multiple sulfation patterns that are recognized by different proteins. For example, HGF binds to domains that have a high level of 6-O-S, while FGF-2 binds to domains where the principal sulfation is by N-S and 2-O-S [30]. The number of proteins that express heparan binding domains is limited. However, many of these molecules are involved in the regulation of angiogenesis and are present in adipose tissues (Appendix A).

As HSPGs enter the extracellular regions, they become subject to modification by other sets of cell-derived enzymes that modify the structure and sulfation of HS chains. Sulfatases are a family of enzymes that mediate the selective removal of sulfate groups from HS chains [26]. Two heparanases mediate the cleavage of specific carbohydrate linkages, thus releasing fragments of the HS chain [31]. Individuals with type I diabetes exhibit high levels of heparanases activity resulting in poorly sulfated HS associated with β-cells [32]. The fragmentation of HS chains releases chemokines that recruit inflammatory cells that destroy the β-cells. Similar events occur in mature adipose tissues exposed to elevated levels of circulating glucose. This results in the upregulation of heparanase production by adipocytes and other cells in the tissue. Additionally, elevated glucose activates inflammatory cells resulting in inflamed adipose tissues characteristic of obese individuals. These enzymes become important in diabetic individuals as the size and sulfation of HS are modified by glucose levels [31]. The HS-modifying enzymes found in adipose tissues are the same as found elsewhere; however, it is possible that their regulation is modified.

Wilsie and others, using the 3T3-L1 adipogenic model, demonstrated that cell surface HSPGs play a major role in differentiating adipocytes from pre-adipocytes. They found an increase in the presence of sulfated PGs upon induction of the 3T3-L1 cells [33]. However, when the cultures were treated with xyloside compounds, a reduction in sulfation was observed and was accompanied by reduced lipid uptake by the cells. The xylosides compete with HS and CS chains for their attachment to their core proteins, resulting in a low sulfated PG [34]. Other studies have shown that apolipoprotein E-enriched very low-density lipid (VLDL) possesses heparan-binding domains that allow this molecule to be concentrated on adipocyte surfaces [33,35]. Adipocytes also produce and secrete lipoprotein lipase (LPL), which also associates with HS through its heparan-binding domain [33,35]. The mechanism by which sulfate regulates lipid uptake is not completely clear. One proposed possibility is that VLDL and LPL are concentrated in proximity to each other on the cell surface and that LPL mediates the release of triglycerides. These are then internalized by fatty acid transporters. Alternatively, VLDL is concentrated on the surface adjacent to the VLDL receptor and/or the lipoprotein receptor-related protein, which mediate the internalization of lipids [33,36] (Figure 2). This process may also occur when mature adipocytes become hypertrophic. In addition, 3T3-L1 cells grown under high glucose conditions exhibit a reduction in cell surface sulfation and a release of bound LPL [37]. In short, HS plays a role in glucose metabolism.

The heparan-binding protein insulin-like growth factor binding protein-2 has been shown to interact with preadipocyte cell surfaces to inhibit their development. This further emphasizes the importance of HSPGs in the regulation of adipogenesis [38]. One of the adipocyte cell surface HSPGs is glypican-4, a glycosylphosphatidylinositol-anchored molecule [39]. The expression of this PG is higher in visceral adipose tissues than in subcutaneous adipose tissues, and its expression increases with body fat content. One of the functions of glypican-4 is to interact with the insulin receptor to enhance adipocyte differentiation and hypertrophy [39]. The core protein can be cleaved, allowing release from the cell to circulate as an adipokine.

Matsuzawa’s group inhibited HS synthesis in 3T3-L1 cells using CRISPR-Cas9 technology to delete the *Ext1* gene, an enzyme involved in HS synthesis and found that this loss resulted in reduced glucose uptake and insulin-dependent intracellular signaling via the BMP4-FGF1 pathways [40]. They further developed mutant mice in which HS chain synthesis was partially inhibited and found a reduction in visceral adipose tissues. This group also demonstrated the role of HS in regulating insulin secretion by pancreatic β-cells [32,41]. Thus, HS plays a dynamic role in multiple organs to regulate glucose levels.

Cell surface HSPGs also act as co-receptors for some growth factors. Fibroblast growth factor-2 and hepatocyte growth factor, both considered adipokines, bind to HS, which then presents these molecules in an active manner to the appropriate cell surface receptors to initiate signal cascades [42]. These factors combine with other HB-binding factors to regulate angiogenesis in adipose tissues: these include vascular endothelial growth factors (VEGF), platelet-derived growth factor (PDGF), insulin-like growth factor (IGF), transforming growth factor- β (TGF- β), and angiopoitin like protein 4 [43]. HB-EGF is produced by multiple cells in adipose tissues: macrophages, vascular endothelial cells, and adipocytes. Its expression is upregulated by oxidative stress and obesity. One of its functions is to increase lipoprotein production by the liver [44].

Heparan-binding EGF is produced in a variety of cells and organs as a transmembrane-bound molecule that can be released by protease cleavage. In adipose tissues, it is produced by adipocytes and pro-inflammatory macrophages [41,45]. It circulates and concentrates in the vasculature, particularly in the liver, where it increases lipoprotein production. It also activates the EGFR and ERB4 and induces the production of oxidants. Importantly, it has been shown to be upregulated in obesity [44].

Adiponectin is produced preferentially by subcutaneous adipocytes and is released into the circulation of lean versus obese individuals, where it promotes insulin sensitivity [46,47]. Both pro- and anti-angiogenic properties have been ascribed to this factor [6]. This adipokine does not possess an HB-binding domain, but it aggregates with PDGF-BB that does associate with HS [48]. This association indicates that it tends to concentrate in subendothelial regions where it may interfere with the activities of PDGF.

Leptin, a product of adipocytes, is released into the circulation, where it travels to the hypothalamus to regulate satiety [3]. It is more highly produced in obese individuals than in lean individuals. Leptin also plays a role in adipose tissues to promote the formation of fenestrated microvessels. In this regard, it acts in a similar manner to VEGF-A, which at high concentrations also promotes leaky vessels [5].

Adipose tissues are highly vascularized, and this feature is modulated as these tissues expand or contract. Adipocytes produce angiogenic factors such as VEGF-A_165_, HGF, PDGF-BB, and ANGPL-4 play roles in vascular expansion or contraction [6,42]. All these molecules contain HB-binding domains. As such, these and other HB-binding molecules become concentrated and protected by their presence in BMs. BMs release these molecules gradually, creating gradients [49] that are necessary for development events such as angiogenesis (Figure 3). Perlecan has been shown to regulate neo-angiogenesis by modifying the concentration of VEGF and its interaction with the VEGF receptor-2 on vascular endothelial cells [50]. The controlled release of angiogenic factors such as FGF-2 plays a role in adipose tissue engineering [51,52]. These molecules are also released from adipose tissues and can also be considered adipokines [4].

HSPGs in BMs play critical roles in the sequestration, concentration, and protection of heparan-binding factors [25]. Furthermore, these structures regulate the diffusion of these molecules, thus creating gradients important for cellular communication and guidance in migration. One area where diffusion gradients are critical is angiogenesis. Molecules like VEGF-a, PDGF, and HGF play critical roles [50]. Adipose tissues undergo constant modification. Both pro- and anti-angiogenic factors play roles in these processes [6].

## 5. In Vitro Adipogenesis

Cell culture has been extensively employed to study adipocyte development. Early studies concentrated on the use of murine 3T3-L1 cells [54]. However, human adipose stem cells (ASCs) and mesenchymal stem cells (MSCs) have more recently been shown to develop as adipocytes following induction [16]. With the identification and isolation of human adipose stem cells, it has become possible to generate more complex organ-like cultures. All of these populations, when cultured in an adipocyte induction medium, undergo a differentiation profile very similar to that seen in vivo. Laminin is an early secreted product and is followed by other BM molecules [15,16,55]. Each adipocyte in the culture is surrounded by a BM upon completion of the induction process. Induced cells acquire the CD146 antigen early upon induction. This provides a marker to separate responsive versus non-responsive cells.

## 6. Tissue Engineered Adipose Tissue

Understanding adipose tissue functions have relied largely upon genetic studies in mice [1]. However, mice differ in a number of critical respects from humans. Repetition of these genetic studies in humans is not feasible. An alternate approach is to develop in vitro assays that can be used for these studies.

The ability to tissue engineer complex, vascularized human adipose tissue has been demonstrated in multiple laboratories [16,56,57,58,59,60,61,62,63]. Differentiated adipocytes are amitotic and not readily adaptable for such constructs. Adipose tissues contain adipocyte stem cells that can be activated during hypertrophy of these tissues. As such, a primary source for these stem cells is the stromal vascular fraction (SVF) from adult adipose tissues [64,65]. The excised tissue is digested with a cocktail of enzymes that release individual cells. This cellular mixture is subjected to mild centrifugation, during which the lipid-containing adipocytes float while the remaining cells form a pellet. This pellet fraction contains a complex mixture of cells that is termed the SVF. The fraction contains stromal cells, stem cells, pre-adipocytes, vascular cells, macrophages, and other immune cells. Adipocyte stem cells are removed from this mixture based on their cell surface antigen profile.

Tissue-engineered vascularized adipose tissue constructs contain three basic cellular elements: adipocytes, stromal cells, and vascular cells [16]. However, additional cellular types may be added to increase complexity. Freshly excised adipocytes are terminally differentiated cells that do not respond well in culture. Instead, a population of adipocyte stem cells can be obtained from the SVF, or bone marrow are suitable starting populations [66,67]. These cells can be selected, and culture expanded to achieve a large cellular population. Cells from the SVF are better suited as they contain a higher percentage of inducible cells than those from bone marrow. Stromal cells are required to produce an extracellular matrix that holds the various cellular components together. In addition, these cells also release angiogenic factors [53]. They are also essential for their ability to initiate and sustain vascular tubule formation in the constructs [53,68]. They are also essential for their ability to initiate and sustain vascular tubule formation [53].

The cultures now contain sufficient cellular and matrix support for vascular cells to spontaneously migrate and associate to form arrays of vascular tubules. The critical issue here is which population of vascular cells to use for this purpose [65]. Human umbilical vein vascular endothelial cells are commercially available, relatively easy to culture, and adaptable for TE purposes. However, these are large vessel endothelial cells, whereas adipose tissue contains microvascular endothelial cells [6]. Human dermal microvascular endothelial cells are also commercially available but are somewhat more difficult to work with than are HUVECs, but they have been employed in these studies [59,64,69]. Another potential source for microvascular endothelial cells is the SVF [64], which appears to be a more appropriate source for TE adipose tissues as these cells have already been adapted to adipose tissues. Human adipose microvascular endothelial cells currently have limited commercial availability. ScienCell Research Laboratories provides these cells as a catalogue item. Another source for these cells is iXCells Biotechnologies, which will isolate them on a customized basis. The SVF contains all the cells needed to make a vascularized adipose tissue. An alternate approach is to mix SVF cells and allow them to self-assemble to form a vascularized construct [63].

Crosstalk between various stromal vascular cells and adipocytes regulates the endocrine function of WAT. This crosstalk is facilitated by HSPGs that regulate the diffusion of chemokines and signaling factors. The direction that this crosstalk can take depends upon the immunological status of the tissue. Animal studies and traditional cell culture studies cannot fully replicate the varied and changing situations that occur in humans. More complex organoid systems are needed to better emulate the actual in vivo state. Rogal et al. [70] have developed a WAT-on-chip approach that can be modified to replicate various immunological states to study the physiology of this system.

The end result is a vascularized adipose tissue that can be used for in vitro studies, or which can be prepared for in vivo implantation in athymic mice [55,59]. These constructs express BMs around both adipocytes and vasculature and contain the HSPG perlecan (Figure 4). However, it is also possible to modify the basic vascularized adipose construct through the insertion of other cell types typical of adipose tissues. Macrophages and immune cells are also present and play a role in creating inflammatory tissue in obese individuals [5]. It is also possible to modify culture conditions to assess the function of specific components. Aubin et al. [71] added tumor necrosis factor-α to the culture medium and found a modification in the release of adipokines into the culture medium. High glucose levels have been shown to modify the structure of HSPGs through an increase in heparanase production [31]. This creates an environment where heparan-bound factors are released so that they can interact with macrophages and immune cells. There are also culture techniques that can be used to modify the sulfation of HSPGs, as demonstrated by the incorporation of xylosides into the culture medium [33,34]. Other possibilities include the genetic modifications of pre-adipocytes and/or vascular cells to up- or down-regulate various genes that control the sulfation and syntheses the carbohydrate chains for HS. Sulfation of GAG chains is mediated by two families of genes: one family mediates the attachment of sulfates to chondroitin, dermatan, and keratan sulfates, while the second family mediates attachment to heparan sulfates [31]. Such studies would provide needed information to understand better the role of HSPGs in adipose tissue production of adipokines.

Adipose tissues contain multiple cellular components that extend beyond the base elements used to make the engineered tissue and vascularized constructs described above [64]. It is possible to insert other cell types such as macrophages and immune cells to assess their effects and increase the complexity of the constructs to mimic native tissues. Many potential variations in the cell composition of such cultures are feasible. It is also possible to modify culture conditions to test the functions of specific components.

The culture conditions can affect cellular interactions. As indicated above, HSPGs play a major role in the development and function of adipose tissues. The sulfation of the HS chains can be modified in culture. The chains are attached to their core proteins via a xyloside linkage. The addition of xyloside compounds to the medium competes for CS and HS chain attachment [34]. This results in poorly sulfated proteoglycans. Treatment of 3T3-L1 cells with a xyloside compound resulted in the inhibition of lipid uptake [19]. The addition of these compounds to complex culture would provide a means to better understand the role of sulfation in adipose tissue function.

## 7. Discussion/Conclusions

The studies referenced above indicate that HSPGs play an important role in the development and function of adipose tissue. Further, these molecules also have a role in the establishment of inflammation in adipose tissue of diabetic individuals. The functional roles of HSPGs are not limited to adipose tissues as they are ubiquitous [44,72]. Sulfation of HS is critical for function as sulfate groups are non-randomly organized to create codes that are recognized by HB-factors [27]. Many of these factors are present in adipose tissues, where they mediate communication between adipocytes and vascular cells [73]. This is important for the maintenance of adipose tissues as an endocrine organ that distributes a wide variety of adipokines to other regions of the body [4,6]. However, it is also important to realize that HS chains can be modified once they are in the extracellular regions. There are two families of enzymes that are released by cells, sulfatases, and heparanases, that modify HS chains and their functions [28,31]. Elevated blood glucose levels have been shown to significantly upregulate the production of heparanases by different cell types [40]. This releases chemokines in adipose tissues that recruit inflammatory cells.

It is possible to modify HS in vitro as this has been demonstrated by adding xyloside compounds to culture media [33,34]. These compounds compete for the attachment sites on core proteins with the result of producing HSPGs with fewer HS chains, hence decreasing sulfation. The observation that glucose levels have a direct effect on HS chain structure suggests that modifying glucose levels in culture media may also result in the modification of sulfation levels [37,40]. There are also more subtle ways that HS sulfation can be modified. Many genes are associated with the glycosidic production and sulfation of HS chains. It has been shown in other systems that a modification of a single sulfotransferase gene can produce physiologic responses. Axelsson et al. [74] inactivated the 2-O-sulfotransferase gene in mice that induced a modification in HS structure so that 6-O- and N-sulfation were increased. This modification resulted in enhanced neutrophil recruitment from the vasculature and increased tissue inflammation. Hwang et al. [75] examined the role of hypermethylation of the 3-O-sulfotransferase 2 genes in lung cancer and found that reduction of methylation restored the ability of tumor cells to proliferate and migrate. These studies indicate that modification of a single sulfotransferase can have physiological effects. It should be possible to genetically modify adipocytes precursors and/or vascular endothelial cells prior to their incorporation into vascularized adipose tissue construct to determine the effects of adipokine production and support for inflammation.

This means that it is now possible to design experiments where HS and/or its sulfation is modified to better define how adipocytes and vascular endothelial cells communicate with each other. This communication is important for the production and release of adipokines. Other experimental studies could focus on the creation of an inflammatory adipose tissue with the addition of macrophages and other inflammatory cells. Intercellular signaling within adipose tissues plays a major role in their endocrine function. This review has concentrated on the role of HSPGs in mediating adipocyte function and intercellular communication. However, our knowledge of how HS regulates these events is still limited. Thus, more experimental studies are needed.

## Figures and Tables

**Figure 1 biomedicines-10-02115-f001:**
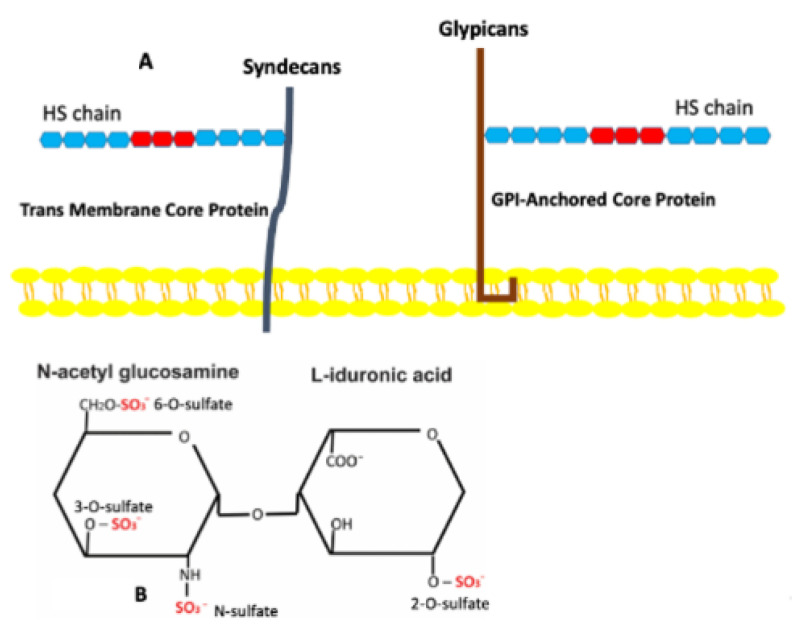
(**A**) Cell Surface HSPGs. HSPGs consist of a core protein to which one or more HS chains are covalently attached. Those HSPGs on the cell surface are integrated into the cell membrane either as transmembrane molecules, such as syndecans or as GPI-anchored molecules, such as glypicans. The HS chains contain domains (in red) in which highly sulfated disaccharides reside. (**B**) Possible Sulfation Sites on HS Disaccharides. There are four possible sulfation sites on HS disaccharides that contain L-iduronic acid.

**Figure 2 biomedicines-10-02115-f002:**
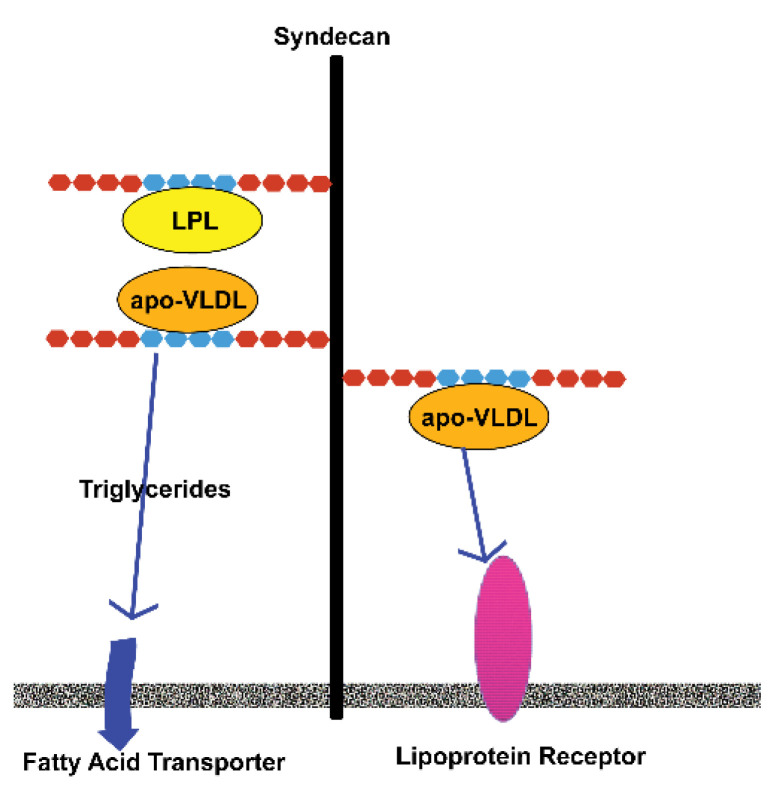
**Cell surface HSPGs promote the uptake of lipids by adipocytes.** Adipocytes produce and secrete lipoprotein lipase (LPL) that becomes concentrated on the adipocyte surface through its interaction with HS. Lipoproteins (apo-VLDL) from the circulation also possess an HB-domain that mediates their attachment to HS on the adipocyte surface. This makes it possible for LPL to mediate the release of triglycerides from the apo-VLDL, and these molecules are internalized by fatty acid transporters. Alternatively, apo-VLDL concentrated on the cell surface by their interactions with HS can be internalized through their binding to lipoprotein receptors. These are possible mechanisms by which adipocytes acquire lipids during their development and maturation. Adapted from the work by Wilsie et al. [33].

**Figure 3 biomedicines-10-02115-f003:**
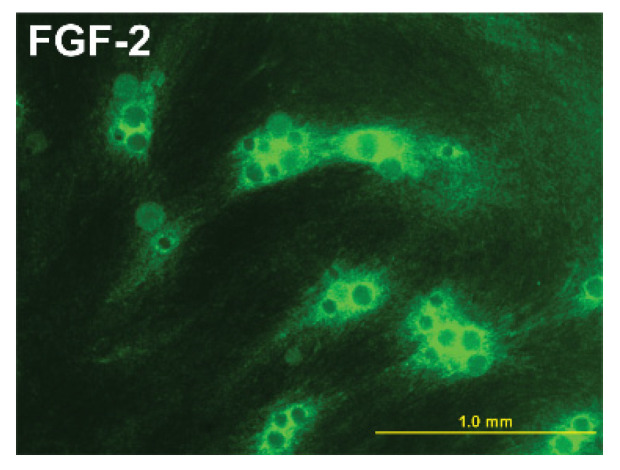
**Concentration and diffusion of HB-binding growth factors.** The human dermal fibroblast was cultured to create a 3D matrix. Agarose beads conjugated with heparin were integrated into this matrix. The cultures were immunostained using an antibody for human FGF-2. As shown in the figure, FGF-2 is highly concentrated around these beads. A gradient develops from these beads into the surrounding cultures. In cultures without beads, FGF-2 cannot be detected by immunohistochemical staining but can be detected in a conditioned culture medium using sensitive ELISAs [53].

**Figure 4 biomedicines-10-02115-f004:**
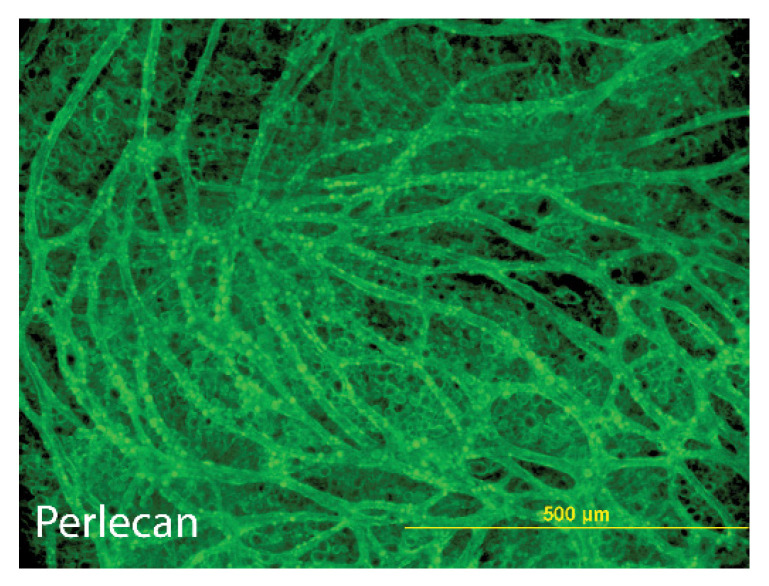
**HSPG perlecan in tissue engineered vascularized adipose constructs.** This is a tissue engineered vascularized adipose construct that was immunostained for the heparan sulfate proteoglycan perlecan. Note that the linear vascular structures and the individual adipocytes are both positive. Stromal cells are present but are not positive for perlecan. Adapted from work referenced previously [16,55].

**Table 1 biomedicines-10-02115-t001:** Heparan Sulfate Proteoglycans [18].

Proteoglycan	Present in Adipose Tissues	Location
	Full-Time HSPGs	
Syndecans 1-4	Syndecans 1, 3, 4	Cell Surface
Glypicans 1-6	Glypican 4	Cell Surface
Perlecan	Yes	Basement Membrane, Matrix
Agrin	Unknown	Matrix
Type XVII Collagen	Unknown	Matrix
	Part-Time HSPGs	
CD44	Yes	Cell Surface
Betaglycan	Yes	Cell Surface

## Data Availability

Not applicable.

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
