# Peer review of "Heparan Sulfate: A Regulator of White Adipocyte Differentiation and of Vascular/Adipocyte Interactions"

_biomedicines, 2022, doi:10.3390/biomedicines10092115_

Round 1
Reviewer 1 Report
This manuscript consists of a short review that will be of interest to all readers working of adipose tissue, with a special focus on the extracellular matrix (ECM) elements that are heparan-sulfate proteoglycans.
It is concise and well structured. It could however benefit from two more figures/tables to really capture the topic and summarize better the findings of this literature review. For example, Section 3 could benefit from an illustration of the structural make up of HSPGs. While figure 1 is instructive, a structural image of the HSPG in their entirety would be helpful so we do not have to refer to another manuscript or review. (thinking of students here that might want to find more info in a single document instead of looking elsewhere).
-Section 3 would also benefit from a table listing HSPG found in (human) AT (are there more than perlecan, syndecan and glypican-4, the 3 named in the text? )
The conclusion mentions the following: “Further, these molecules also have a role in
the establishment of inflammation in adipose tissue of diabetic individuals.”
Indeed, throughout reading the review, the information relating to modulations observed in diabetes is very interesting. It is mentioned also the impact on enzymes in section 4 in relation to sulfation. But what about the molecules themselves, are they different in nature in AT depots of obese or diabetic patients versus healthy adipose tissue (e.g., for perlecan, syndecan, etc.). If so, this information could be part of the new Table suggested above. If no such modulation is seen (or not known), it could be stated as well for clarity.
Line 230: Adipocyte stem cells are removed from this mixture based upon their cell surface antigen profile.
I would rephrase this sentence since the large majority of teams using ASCs for adipose tissue engineering are not enriching the cultures apart from adhesion and cultures, due to the lack of a perfect ASC marker for selection. Not that many teams to my knowledge, are selecting cells by flow cytometry before amplification, at least for human ASCs. It might have been done more extensively in the past for murine cells when combination of markers were searched for? I would rephrase so it does not read like all teams enrich their cultures from the start using flow cytometry-base selection using cell-surface markers.
Line 254: however, they are currently not commercially available. Concerning adipose-tissue derived endothelial cells, it is true that these cells are not widely available and are not likely to be affordable for 3D tissue engineering purpose, even though it should be noted that they could be ordered at https://www.ixcellsbiotech.com/human-primary-cells/human-adipose-microvascular-endothelial-cells-hamec.
And
https://www.sciencellonline.com/human-adipose-microvascular-endothelial-cells.html
Line 338: Other experimental studies could focus on the creation of an inflammatory adipose tissue with the addition of macrophages and other inflammatory cells.
Although smaller in size than other 3D models of vascularized adipose tissue, in 2022 an immunocompetent white adipose tissue on chip was reported: DOI: 10.1002/advs.202104451
Please define HB line 118
Secreted by adipocytes (line 184) are named, by macrophages too, but what about the stromal cells, they secrete too no?
Line 351: Figure 3 are human cells so REB approval should be stated at the end of the manuscript. Are figure 4 human cells too? Please cite the source of the cells shown.
Section 5 please define ASC and MSC at first occurrence
Line 91 mentions a supplementary Table 1 that could not be found.
Author Response
We wish to thank the reviewer the many helpful comments and suggestions, which were the basis for the revisions of the manuscript. As a result, substantial sections of the text were rewritten. New text appears in red font.
Section 3: The text was substantially revised and a new figure was placed in the text. This figure shows the basic HSPG structure and also incorporates the previous figure 1. Also, a table listing HSPGs and their association with adipose tissues was also inserted into the text.
Information regarding WAT-on-chip was included as were commercial sources for WAT microvascular endothelial cells. Information regarding the sources for cellular populations used for figures 3 and 4 was included.
Reviewer 2 Report
The aim of this review article was to characterize the role of heparan sulfates in biology of preadipocytes and mature white fat cells. In my opinion manuscript is interesting, nevertheless the Authors should consider to improve the structure (especially titles of sections) of the manuscript because in the present form manuscript seems to be not really informative. These are my comments to the Authors:
1. The title of section 4 is not really informative. In addition to sulfation of heparan sulfates there are information regarding the role of heparan sulfates in controlling many several aspects of fat tissue physiology. These points should be stated as separate chapters. The percent structure make the manuscript not really informative and unclear.
2. In Vitro Adipogenesis. What is the reason of this chapter? In my opinion in this part of the manuscript the Authors should characterize what we know about the role of heparan sulfate in controlling white preadipocytes proliferation and their differentiation into mature fat cells.
3. Section “Tissue Engineered Adipose Tissue” – in my opinion first part of this chapter is a bit out of scope of this manuscript and can be shorten by the Authors.
4. I would suggest to add a figure summarizing current knowledge about the role of heparan sulfates in modulating fat tissue physiology.
5. Some sentences seam to lack references.
Author Response
The title to section 4 was changed. This section is meant to be a brief review of sulfation and how this affects the metabolic function of adipose tissues.
Tissue Engineered Adipose Tissues. Much of our current knowledge of adipose tissue biology is based upon genetic studies in mice. While there is much similarity to mice and humans, there are also significant differences. Similar genetic studies in humans are not feasible. An alternative approach is needed. Tissue engineered adipose tissue provide this alternative. This makes this topic relevant to this manuscript. Also reference to WAT-on-chip has been included in this section as an alternate approach to these studies.
There are multiple sentences that rely upon a common set of references. All information provided was referenced
Reviewer 3 Report
Heparan Sulfate: A Regulator of White Adipocyte Differentiation and of vascular/Adipocyte Interactions
Dear Authors,
Congratulations on writing such an interesting study. This manuscript aims to reveal the role of HSPGs in mediating adipocyte function and intercellular communication.
The following are my comments and suggestions:
Abstract and Introduction:
- My most of comments are on the presentation of information. The manuscript needs to be edited for pictorial presentation. For example, kindly provide the diagrams for diabetic condition establishment and how Heparan Sulfate involves there.
- HS relationship with type 2 and type 1 diabetes. What’s the distinct feature in diabetes 2 and 1 with respect to HS involvement?
- Kindly provide what happens with HS chin in case of prolonged starvation?
- What will happen in the case of a ketogenic diet? Any changes induced by ketone bodies?
Discussion:
So, based on these conclusions, how do we proceed with managing hyperglycemia in the diabetic subjects?

Author Response
In relationship of HS to type 1 diabetes. See reference 32.
Also, please see figure 2.
There is very sparse data regarding HSPG during starvation and ketone production. This is why it is important to develop in vitro applications to study such issues in humans.
We hope that you will find this a suitable contribution to your Special Issue.
Round 2
Reviewer 2 Report
Authors addressed my comments adequately.